# Regional Disparities in HPV Vaccination Coverage Among Girls Aged 9 to 14 Years in Togo: Lessons Learned from the Recent Supplementary Immunization Activities

**DOI:** 10.3390/vaccines13040373

**Published:** 2025-03-31

**Authors:** Dadja Essoya Landoh, Issifou Yaya, Amevegbe Boko, Kodjovi Adjeoda, Yaovi Temfan Toke, Adidja Amani, Yerima Mouhoudine, Ado Mpia Bwaka, Nsiari-Mueyi Joseph Biey, Charles Shey Wiysonge, Franck Fortune Roland Mboussou, Hèzouwè Looky-Djobo, Tsidi Agbeko Tamekloe, Toyi Nyulelen Mangbassim, Tchasso Kenao, Amadou Bailo Diallo, Fatoumata Binta Tidiane Diallo, Benido Impouma, Ann Lindstrand, Marin Kokou Wotobe, Didier Koumavi Ekouevi

**Affiliations:** 1World Health Organization, Country Office, Lomé BP 1504, Togo; tamekloet@who.int (T.A.T.); dialloa@who.int (A.B.D.); boussokadi@yahoo.fr (F.B.T.D.); 2Patient-Reported Outcomes Research (PROQOL), Health Economics Clinical Trial Unit (URC-ECO), Hotel-Dieu Hospital, AP-HP, 75004 Paris, France; iyayad@yahoo.fr; 3CHU-SO, Unité de Recherche en Santé des Populations (URESAP), Épidemiologie des Maladies Infectieuses et Sciences Sociales (Épidé-MISS), Lomé BP 1515, Togo; 4ECEVE, UMR-S 1123, Paris Cité University, Inserm, 75006 Paris, France; 5Division de l’Immunisation, Ministère de la Santé, Lomé BP 386, Togo; amevegbe12@yahoo.fr (A.B.); adjtg@yahoo.fr (K.A.); lookydjobo@yahoo.fr (H.L.-D.); 6Fonds des Nations Unies pour L’enfance (UNICEF), Bureau Togo, Lomé BP 4042, Togo; yawovitemfan@gmail.com (Y.T.T.); machristian2008@yahoo.fr (T.N.M.); tkenao@unicef.org (T.K.); 7World Health Organization, Regional Office for Africa, Brazzaville P.O. Box 06, Congo; amaniad@who.int (A.A.); sheyc@who.int (C.S.W.); impoumab@who.int (B.I.); 8Ministère de la Santé, Lomé BP 386, Togo; dineyerima@yahoo.fr (Y.M.); wotobemarin@yahoo.fr (M.K.W.); 9World Health Organization, IST West Africa, Ouagadougou P.O. Box 7019, Burkina Faso; bwakaa@who.int (A.M.B.); bieyj@who.int (N.-M.J.B.); 10World Health Organization, IST Central Africa, Libreville, Gabon; mboussouf@who.int; 11Department of Immunization, Vaccines and Biologicals (IVB), World Health Organization, 1211 Geneva, Switzerland; lindstranda@who.int; 12Centre de Formation et de Recherche en Santé Publique, Université de Lomé, Lomé BP 1515, Togo; didier.ekouevi@gmail.com

**Keywords:** HPV vaccination, girls, supplementary immunization activities, coverage, disparities, Togo

## Abstract

Background/Objectives: Human papillomavirus (HPV) vaccination is a critical intervention to prevent cervical cancer, especially in settings where screening is limited. In Togo, cervical cancer is the second most common cancer in women. Togo conducted an HPV vaccination campaign for girls aged 9–14 from 27 November to 1 December 2023, followed by introduction of the vaccine into routine immunization. This study aimed to assess regional disparities in vaccination coverage during this campaign. Methods: A cross-sectional study was conducted using data from the nationwide HPV vaccination campaign. The target population included girls aged 9–14, following school and community-based enumeration. The campaign employed school-based, health facility-based, and community-based vaccination strategies. Data were collected via multiple tools, and monitoring was carried out through daily reports and digital tracking. Results: Out of the estimated 654,402 eligible girls, 304,457 (46.5%) were vaccinated. Vaccine coverage varied significantly by region, ranging from 76% in Kara to 15% in Grand Lomé. In-school settings accounted for 91.3% of vaccinations, with the fixed strategy covering 55.4%. In total, 11 districts exceeded 80% vaccine coverage, while 15 districts had <50%. The highest rate of adverse events following immunization was observed in the Maritime region, primarily involving minor symptoms. Conclusion: Although progress was made in HPV vaccination coverage in Togo, regional disparities highlight the need for targeted interventions. Strategies such as expanding vaccine access, enhancing awareness campaigns, and integrating HPV vaccination into routine immunization could improve coverage. Addressing logistical and cultural barriers is also crucial for equitable vaccination, aiming to achieve international benchmarks and reduce HPV-related disease burdens. Further research should explore qualitative factors influencing vaccine acceptance.

## 1. Introduction

Human papillomavirus (HPV) is a common virus that encompasses over 200 distinct types, categorized into five primary genera: Alpha, Beta, Gamma, Mu, and Nu [1]. These HPV types are typically classified as either low-risk (LR—non-carcinogenic) or high-risk (HR—carcinogenic) [2]. Persistent infection with high-risk strains of HPV, particularly types 16 and 18, is strongly associated with the development of cervical cancer. Additionally, HPV infection has been implicated in the development of other cancers, including those of the anus, penis, vagina, vulva, and oropharynx [3].

More than half a million (604,000) new cases of cervical cancer are reported worldwide in 2020, of which 342,000 deaths occurred, and approximately 90% of these deaths occurred in low- and middle-income countries, including Sub-Saharan Africa [4], where the estimated pooled prevalence of HPV among women is 32.3% [5]. This region bears one of the highest global burdens of HPV, which is reflected in high cervical cancer incidence rates [6].

HPV vaccination is crucial in reducing the global cervical cancer burden, especially in developing countries where effective screening and treatment are lacking, making vaccination particularly important [7]. Studies show that widespread vaccination can reduce cervical cancer incidence by nearly 90% among young girls vaccinated before the age of 17 years, with a single dose providing 97.5% protection against persistent infections with HPV types 16 and 18 [8,9]. These findings highlight the critical role of high HPV vaccination coverage, particularly in resource-limited settings, in significantly lowering cervical cancer rates and protecting women’s health.

Primary prevention of cervical cancer is essentially based on HPV vaccination and healthy lifestyles [10]. Three vaccine types are currently available and marketed in many countries around the world to prevent HPV related disease: the bivalent vaccines (against genotype 16 and 18), the quadrivalent vaccine (against genotype 6, 11, 16, and 18) and the nonavalent vaccine (against genotypes 6, 11, 16, 18, 31, 33, 45, 52, and 58). All three vaccines include the oncogenic genotypes 16 and 18 that cause most of the cervical cancer [11]. HPV vaccination has been shown to be cost-efficient in low-income countries, including those of Sub-Saharan Africa [12,13].

Overall, raising awareness about HPV and its link to cancer, promoting vaccination, and ensuring access to screening services are essential strategies for reducing the burden of HPV-related cancers [14].

In Togo, cervical cancer ranks as the third most common cancer in terms of incidence (19.1 cases per 100,000 individuals) in both sexes, following prostate cancer (32 cases per 100,000) and breast cancer (30.7 cases per 100,000). Among women, cervical cancer is the second most common cancer (15.7%) after breast cancer (29.4%), based on the GLOBOCAN 2020 global estimates [15]. To mitigate the morbidity and mortality associated with cervical cancer, Togo conducted a nationwide catch-up vaccination campaign with the human papillomavirus (HPV) vaccine for girls aged 9 to 14 years from 27 November to 1 December 2023. Following this, on 4 December 2023, the HPV vaccine was introduced into the routine immunization schedule for 9-year-old girls. Togo conducted a demonstration of the HPV vaccine introduction, covering a cohort of 15,272 girls aged 10 years who received two doses of HPV Vaccine in 2015 and 2016 in two health districts (Tchamba and Golfe) [16].

Since the launch of the Vaccine Alliance (Gavi) program in the country, Togo has consistently achieved over 90% vaccine coverage for most antigens under the Expanded Program on Immunization [17,18], despite the socio-cultural challenges that could have favored vaccination hesitancy in some administrative regions. From the scientific literature review, it appeared that vaccine acceptance can be influenced by several factors, including myths and misconceptions about vaccination and knowledge gaps about immunization. In many low- and middle-income countries, as in Togo, there may be challenges to achieving high vaccination coverage rates, such as limited access to healthcare services, vaccine supply chain issues, and vaccine hesitancy among some communities.

In Togo, disparities in vaccination coverage can exist across regions due to various factors, such as access to healthcare services, socioeconomic status, and cultural beliefs [17,18]. These regional disparities may be particularly pronounced due to the country’s geographical and socio-economic diversity. Factors such as urbanization, infrastructure development, and availability of healthcare facilities can significantly impact vaccination coverage rates in different regions.

This study aimed at analyzing the regional disparities in the HPV vaccination coverage in Togo, during the national vaccination campaign held in November 2023. Understanding these disparities is crucial for designing targeted interventions to improve HPV vaccination coverage and reduce the burden of HPV-related diseases in Togo. Research on regional disparities in HPV vaccination coverage can provide insights into the underlying factors contributing to these disparities and inform strategies to address them effectively.

## 2. Materials and Methods

### 2.1. Study Design

This study is a cross-sectional study using data from the recent HPV vaccination campaign targeting girls aged 9 to 14 years. The campaign was implemented nationwide from 27 November to 1 December 2023, aiming to vaccinate at least 90% of the target population with the Cervarix^TM^ bivalent HPV vaccine (GlaxoSmithKline (GSK) in Rixensart, Belgium).

### 2.2. Target Population and Sampling

The target population for the vaccination campaign was adolescent girls aged 9 to 14 years. Before the campaign, a census was conducted by district teams to enumerate all eligible girls across the country. The enumeration process varied by region. In districts with high net school enrollment rates for girls (above 95%), school-based enumeration was through collaboration with school heads and teachers. In regions like Savanes and Kara, where the net enrollment rate for girls is around 80%, in addition to school-based enumeration, community-based enumeration was carried out by community health workers (CHWs) to ensure that out-of-school girls were also counted.

Given the high school enrollment rate in Togo (93.1% for girls in the 2021–2022 academic year), the majority of the target population was expected to be enrolled in school, with a comprehensive strategy to reach both in-school and out-of-school girls.

### 2.3. Vaccination Campaign Strategies

The campaign was conducted using three main settings to administer the HPV vaccine:-School-based vaccination: the vaccination was administered directly in schools to reach most girls who were enrolled in educational institutions, including public, private, and Koranic schools.-Health facility-based vaccination: out-of-school girls, as well as in-school girls who missed their vaccination at school, were vaccinated at designated health facilities.-Community-based vaccination: in cases where girls could not be vaccinated at schools or health facilities, mobile vaccination teams were deployed in the communities. This approach involved outreach vaccination strategies to reach out-of-school girls in remote areas and ensure that no one was left behind.

These strategies were supported by a schedule of visits to public and private schools, including Koranic schools. A high school enrollment rate among girls was a crucial factor in the campaign’s planning and implementation. Overall, fixed, outreach and mobile strategies were used in the three setting to cover a maximum of the target population.

### 2.4. Data Collection Tools and Procedures

To ensure systematic data collection during the campaign, a range of tools were developed and deployed, including the following: enumeration and vaccination form (for collecting data on vaccinated girls), out-of-school girls survey form (to identify and record out-of-school girls eligible for vaccination), school survey form (to track in-school girls and vaccination progress), household convenience survey form (for rapid assessment of households), vaccination card (to document each girl’s vaccination status), summary of vaccination results form (to consolidate data at the end of each day).

Additionally, several training and guidance documents were prepared for the vaccination teams, such as training manuals for supervisors and vaccinators, and protocols on detecting, reporting, and managing potential adverse events following immunization (AEFI). Training sessions and micro-planning activities were conducted a month prior to the campaign, involving health and education stakeholders at various levels (regional health directors, district health workers, and education staff). Partners such as the WHO, UNICEF, and Gavi consultants also contributed to these preparations’ activities.

The bivalent vaccine (Cervarix^TM^ GlaxoSmithKline (GSK) in Rixensart, Belgium)., with 672,200 doses for the campaign and other supplies, were received in July 2023, several months before the start of the campaign, and stored according to standard procedures at the national level. A week before the launch of the campaign, the vaccine (625,522 doses), injection equipment, and other supplies were pre-positioned in the health facilities by the district’s teams.

The vaccination teams comprised four members each: two vaccinators (one tasked with collecting the vaccine and the other with its administration) and two recorders (one verifying the age of the recipients, completing vaccination cards and enumeration sheets, and the other filling out tally sheets and ensuring their accuracy). Throughout the vaccination sessions, the vaccinators informed the girls about the targeted disease, vaccination objectives, and common side effects, and advised them to seek medical care in case of adverse reactions. Additionally, each vaccination team conducted awareness sessions on menstrual hygiene and puberty management during school visits.

The Cervarix vaccine was administered intramuscularly in the deltoid region of the left arm. The dose administered per girl aged 9 to 14 was 0.5 mL.

### 2.5. Monitoring and Quality Control

A robust monitoring system was established to track daily progress at both the operational and national levels. Data were collected through various forms, including enumeration sheets, tally sheets from immunization teams, and waste management sheets. These were submitted to health facility managers, who then reported daily figures to district health authorities. The Division of Immunization compiled these data and produced daily immunization bulletins, with support from partners.

For real-time monitoring, data were entered into the District Health Information System 2 (DHIS2), with technical support from the Health Information Systems Program (HISP). Personal data were not collected, and all variables were configured to facilitate analysis.

At the end of each day, coordination meetings were held at districts and region level to review progress and address any challenges encountered during the vaccination sessions.

### 2.6. Data Management and Analysis

Once the vaccination campaign was completed, all data collected during the campaign were transferred to EPI-Info 7.2 software for analysis. Descriptive statistical analyses were performed to assess the coverage, identify gaps, and evaluate the overall success of the campaign.

Virtual debriefing meetings were organized with stakeholders from all districts and regions to discuss the campaign’s results, identify key findings, and strategize for reaching girls who were missed during the campaign.

### 2.7. Ethical Considerations

The study followed ethical guidelines by not collecting any personal data during the campaign. The emphasis was placed on maintaining the privacy and confidentiality of the girls involved. All participants, including the girls and their guardians, were fully informed about the purpose of the vaccination, potential side effects, and advised to seek medical assistance in case of adverse reactions. Additionally, awareness-raising sessions on menstrual hygiene and puberty management were integrated into the vaccination visits, further supporting the girls’ health and well-being beyond the HPV vaccination. An oral consent was obtained from the parents prior to the vaccination.

The Supplementary Immunization Activities (SIA) using Cervarix was recommended by the National Immunization Technical Advisory Group (NITAG) [19] and approved by the Inter agency Coordination Committee (ICC)

## 3. Results

According to the census, the number of girls aged 9 to 14 expected to be vaccinated against HPV was estimated at 654,402 (111,787 girls of 9 years and 542,615 aged from 10 to 14 years), including 179,969 in Grand Lomé Region, 104,442 in Maritime, 141,842 in Plateaux, 63,910 in Centrale, 79,655 in Kara, and 84,583 in Savanes Region.

A total of 304,457 girls (62,284 girls aged 9 and 242,173 girls aged 10–14) were vaccinated out of the 654,402 girls expected (Table 1), with overall administrative coverage of 46.5% (56% for girls aged 9 and 45% for girls aged 10–14).

Vaccination coverage for girls aged 9–14 varies from 76% (Kara region) to 15% (Grand Lomé region). Figure 1 shows administrative vaccination coverage by region.

The median HPV vaccination coverage was 61.6% (IQR: 40.7–80.2%), with great variability within and between regions. The disaggregated vaccine coverage by district showed that Dankpen district had the highest administrative coverage (100%) and Golfe district the lowest (13%). Moreover, 11 districts out of the 39 (28%) exceeded 80% vaccination coverage: Plateaux region (Anié and Est-Mono); Central region (Blitta and Sotouboua); Kara region (Bassar, Dankpen, and Doufelgou) and Savanes region (Kpendjal, Kpendjal-Ouest, Oti, Oti-Sud). A total of 15 districts out of 39 (38%) did not achieve 50% administrative coverage: Grand Lomé region (all districts: Agoè and Golfe); Maritime region (all districts: Avé, Bas-Mono, Lacs, Vo, Yoto, Zio); Plateaux region (Danyi, Haho, Kloto, Kpélé, Moyen-Mono, Ogou, Wawa); Central region (Tchaoudjo). Figure 2 shows vaccine coverage disaggregated by district, while Figure 3 highlights the performance of the districts. The details are displayed in Appendix A

Apart from the school environment, the target group was found and vaccinated in out-of-school settings. A total of 26,353 girls were vaccinated in out-of-school settings (8.7% of the total vaccinated) and 278,104 in schools (91.3%). In addition, 55.4% of the girls were vaccinated through the fixed strategy, while 4.2% during the mobile strategy (Table 2).

Vaccination coverage for girls in and out of school is broadly in line with school enrollment rates by region (Grand Lomé = 98.1%; Maritime 96.8%; Plateaux = 95.7%; Centrale 93.0%; Kara = 91.6%, Savanes = 80.4%), although there are some regions having low coverage rates recorded.

Concerning the AEFI, the highest rate of notification (145.5/100,000 doses) was recorded in the Maritime region. For minor AEFI, the most frequently reported signs were injection site reaction (pain, redness, swelling, which accounted for 86.8%); fever (19.3%) and headache (17.6%). A total of seven serious AEFI cases were recorded throughout the country, including three in the Kara region, and were investigated, with positive outcomes.

## 4. Discussion

This research focused on assessing HPV vaccination coverage in Togo, following a nationwide campaign that targeted girls aged 9 to 14 years old. We sought to identify discrepancies in vaccine uptake among vulnerable population groups. The quantitative analysis revealed that approximately half of the target population received the HPV vaccine, with a similar coverage for both in-school and out-of-school girls. This level of coverage, though significant, highlights gaps when compared to global standards, such as those found in high-income countries. Additionally, regional disparities were observed, particularly a south–north gradient, with socioeconomic factors potentially influencing access to vaccines.

Prior to the implementation of the campaign, the readiness assessment tool was used to assess key components of campaign preparedness, including cold chain management, staff training, vaccine supply, AEFI monitoring, community awareness, coordination, and resource mobilization. This systematic approach helps identify gaps early and implement corrective actions in all districts. All the districts received similar support during preparation and implementation of the campaign.

The HPV vaccination coverage observed in Togo is comparable to findings from a similar study in Tanzania, which reported a 49% coverage rate [20]. However, these results contrast sharply with a systematic review and meta-analysis of school-aged adolescent girls in sub-Saharan Africa, which indicated a significantly lower uptake rate of 28.53% [21]. When compared to high-income countries like the USA (62.8%), Australia (83%) [22], and Scotland (94.4%) [23], Togo’s HPV vaccination coverage lags substantially. This discrepancy likely stems from differences in healthcare infrastructure, vaccine availability, and routine immunization practices between low-resource and developed countries.

The study showed notable regional variations in HPV vaccination coverage within Togo, with a clear south-to-north gradient. Surprisingly, the southern regions, including Lomé, which have higher literacy rates and socioeconomic status, showed lower vaccination coverage compared to some northern regions. This suggests that factors beyond education and economic status may play a significant role in vaccine uptake. Such factors may include awareness, effectiveness, and involvement of local communities during the vaccination campaign. Also, within a region, there are vaccine coverage variations between districts. Most districts with low vaccine coverage are urban, with high impact from misinformation circulating in social medias. The observed disparities highlight the need for region-specific interventions to ensure equitable uptake of the HPV vaccine across the country. We found that the HPV vaccine coverage was 104% in Dankpen district. This discrepancy could be explained by possible inaccuracy in estimating the target population during enumeration process, and data recording errors during the implementation of the supplementary immunization activities. The HPV vaccine coverage rates were consistently higher among 9-year-old girls compared to those aged 10–14 across all regions. This could be attributed to a higher refusal rate among the older age group. Nine-year-old girls, who are most often in primary school, encountered less resistance than their older counterparts in secondary school. Additionally, girls aged 10–14 may have greater exposure to rumors on social media, which could contribute to higher refusal rates.

Several barriers may contribute to the low HPV vaccination rate in Togo [17,18]. Firstly, this campaign was implemented in a context where rumors about the HPV vaccine were widespread on social media, particularly in the southern regions. This led to significant cases of vaccine refusal, especially in these southern regions. Secondly, concerns about side effects, fueled by misinformation, continue to affect vaccine acceptance in Togo. Misconceptions such as the false belief that the HPV vaccine causes infertility or promotes promiscuity are prevalent in some communities, undermining trust in the vaccination. Thirdly, low awareness may also have played a role in the low coverage. In fact, a lack of public understanding of HPV, its risks, and the benefit of vaccination is a major barrier in Togo. This could be explained partly by the limited education and awareness campaigns on the importance of HPV. The absence of comprehensive information prevents individuals from recognizing the importance of vaccination in preventing cervical cancer.

To address these barriers and increase HPV vaccination coverage across Togo, a multi-pronged approach is necessary. Key strategies include vaccine access expansion, by improving healthcare infrastructure and logistics to ensure broader availability of vaccines, particularly in underserved regions. Awareness raising by implementing targeted education campaigns and community dialogues to dispel myths about the vaccine and provide accurate information about the safety of HPV vaccine and its efficacy could also improve coverage. These campaigns should focus on promoting the benefits of vaccination for preventing HPV-related diseases. In addition, healthcare providers and community leaders should use evidence-based communication strategies to counter misinformation and build trust in vaccination programs. Furthermore, HPV vaccination should be incorporated into routine immunization schedules and financial barriers reduced through free vaccination programs. Finally, government agencies, healthcare providers, community organizations, and international partners should work together to overcome logistical, financial, and cultural barriers to vaccination.

### Limitations

While this study provides valuable insights into HPV vaccination coverage in Togo, there are several limitations. First, the study focused primarily on quantitative data, with limited assessment of the qualitative factors that influence vaccine uptake, such as cultural beliefs and community engagement. Furthermore, the study did not account for potential biases in self-reported vaccination data, which could influence the accuracy of the reported coverage rates.

Future research should explore the qualitative aspects of HPV vaccination uptake, including the role of cultural, social, and religious factors in shaping attitudes toward vaccination. Studies that investigate the effectiveness of targeted educational campaigns and intervention strategies in different regions could provide valuable insights for improving vaccine acceptance. Additionally, longitudinal studies tracking vaccination coverage and its impact on cervical cancer incidence would contribute to understanding the long-term outcomes of HPV vaccination programs in low-resource settings.

The findings of this study have important policy implications for improving HPV vaccination coverage in Togo. Policymakers should prioritize efforts to strengthen healthcare infrastructure, particularly in rural and difficult to access areas, to ensure equitable access to vaccination services. Offering free HPV vaccinations could reduce financial barriers and improve uptake among vulnerable populations. Moreover, launching national awareness campaigns that address common misconceptions and promote the benefits of vaccination is critical for increasing public acceptance. Finally, integrating HPV vaccination into the broader national immunization program would ensure its sustainability and long-term success in reducing HPV-related disease burdens. By addressing these policy priorities, Togo can move closer to achieving international benchmarks for HPV vaccination and improving public health outcomes.

## 5. Conclusions

Significant progress has been made in HPV vaccination coverage among girls aged 9 to 14 in Togo; however, disparities persist compared to high-income countries, but also between the different regions. Further interventions and investigations are needed to identify and address barriers to vaccine acceptance, to implement targeted interventions, and to strengthen healthcare systems. Toward these strategies, stakeholders can enhance HPV vaccination coverage, improve equity, and mitigate the impact of HPV-related diseases in vulnerable populations.

## Figures and Tables

**Figure 1 vaccines-13-00373-f001:**
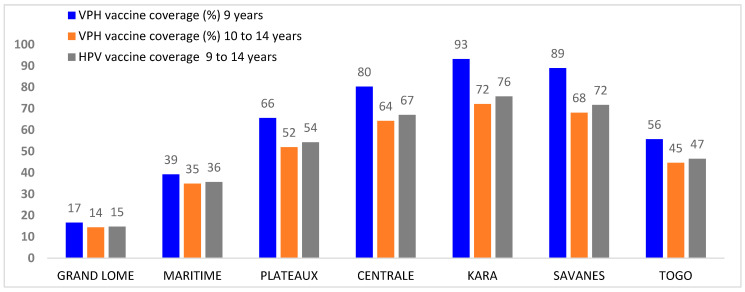
Administrative HPV vaccination coverage by region during SIA for girls aged 9–14 in Togo in December 2023.

**Figure 2 vaccines-13-00373-f002:**
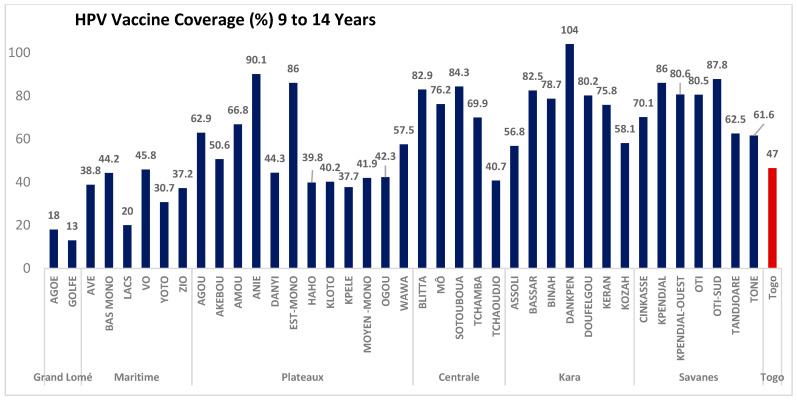
Administrative vaccination coverage by district during the HPV vaccination campaign for girls aged 9–14 in Togo in December 2023.

**Figure 3 vaccines-13-00373-f003:**
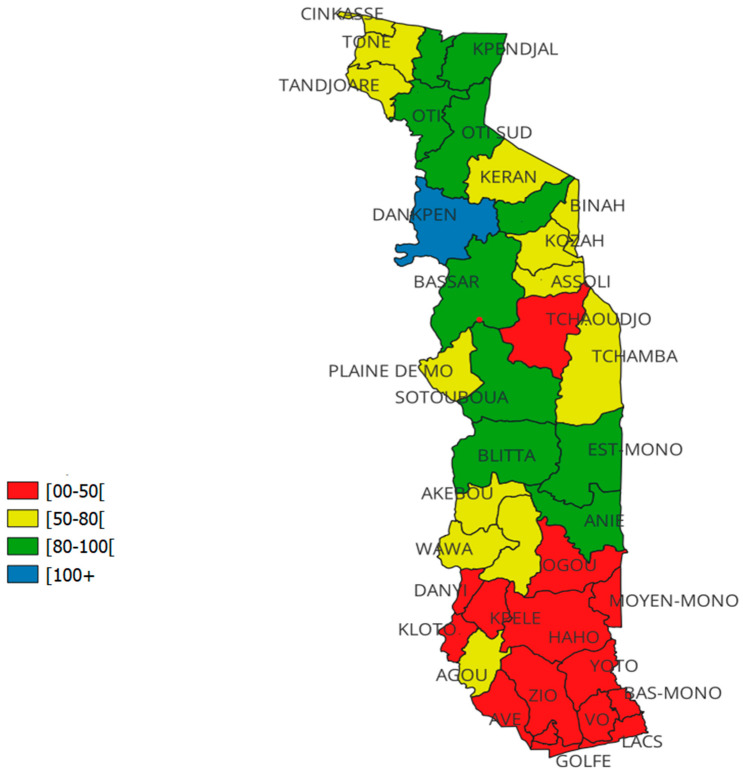
Performance of districts, administrative vaccination coverage during the HPV vaccination campaign for girls aged 9–14 in Togo in December 2023.

**Table 1 vaccines-13-00373-t001:** Number of girls vaccinated against HPV by region, vaccination campaign for girls aged 9 to 14 to prevent cervical cancer in Togo, December 2023.

Region	Expected Number of 9-Year-Old Girls	Expected Number of Girls Aged 10 to 14	Expected Number of Girls Aged 9 to 14	Number of 9-Year-Old Girls Vaccinated	Number of Girls Aged 10 to 14 Vaccinated	Number of Girls Aged 9 to 14 Vaccinated	Number of HPV Vaccine Doses Used	Number of AEFI Reported	
								Serious AEFI	Minor AEFI
Grand Lomé	30,743	149,226	179,969	5092	21,519	26,611	27,068	0	13
Maritime	17,841	86,601	104,442	6998	30,183	37,181	37,673	0	148
Plateaux	24,230	117,612	141,842	15,900	61,030	76,930	77,356	0	185
Centrale	10,917	52,993	63,910	8772	34,034	42,806	43,061	0	43
Kara	13,607	66,048	79,655	12,679	47,644	60,323	60,740	1	33
Savanes	14,449	70,135	84,583	12,843	47,763	60,606	62,572	0	50
Total	111,787	542,615	654,402	62,284	242,173	304,457	308,470	1	472

**Table 2 vaccines-13-00373-t002:** Vaccination coverage by settings, by strategy, and by region during the cervical cancer vaccination campaign for girls aged 9–14 in Togo in December 2023.

Region (Nb of Girls Vaccinated)	Settings		Strategy		
	In School, n (%)	Out of School, n (%)	Fixed, n (%)	Out Reach, n (%)	Mobile, n (%)
Grand Lomé (26,611)	24,790 (93.2)	1821 (6.8)	7561 (28.4)	17,160 (64.5)	1890 (7.1)
Maritime (37,181)	35,961 (96.7)	1220 (3.3)	20,230 (54.4)	15,172 (40.8)	1779 (4.8)
Plateaux (76,930)	70,401 (91.5)	6529 (8.5)	47,176 (61.3)	27,180 (35.3)	2574 (3.3)
Centrale (42,806)	39,646 (92.6)	3160 (7.4)	25,153 (58.8)	16,589 (38.8)	1064 (2.5)
Kara (60,323)	55,913 (92.7)	4410 (7.3)	33,794 (56.0)	23,912 (39.6)	2617 (4.3)
Savanes (60,606)	51,393 (84.8)	9213 (15.2)	34,786 (57.4)	22,887 (37.8)	2933 (4.8)
Togo ( 304,457)	278,104 (91.3)	26,353 (8.7)	168,700 (55.4)	122,900 (40.4)	12,857 (4.2)

## Data Availability

Extracted data are with the authors and at the Ministry of Health and available for sharing on request.

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
