# Peer review of "Regional Disparities in HPV Vaccination Coverage Among Girls Aged 9 to 14 Years in Togo: Lessons Learned from the Recent Supplementary Immunization Activities"

_vaccines, 2025, doi:10.3390/vaccines13040373_

Round 1
Reviewer 1 Report
Comments and Suggestions for Authors
The authors used national HPV vaccine campaign data to calculate vaccine coverages by geographic regions, immunization settings and strategies in Togo. The information should be helpful to improve HPV vaccine coverage in Togo.
Line 95-99, Was this the first HPV campaign in Togo? Was HPV vaccine available prior to Nov. 27, 2023? If so, how many girls aged 9-14 were vaccinated prior to this campaign?
Line 140, Please further clarify if fixed, outreach and mobile strategies were used in all three settings (i.e., school, health and community).
Line 230, What might be the reasons that the HPV vaccine coverage rates were consistently higher for girls aged 9 compared to those aged 10-14 in all regions?
Line 243, What might be the reasons for high variation between districts within a region? For example, the vaccination rates vary from 37.7 to 90.1 in Plateaux.
Line 268, Did cervical cancer burden vary greatly by geographic areas in Togo? How did vaccine coverage compare to the cervical cancer incidence in the area? Was the vaccine acceptance rate higher in the region/district with higher cervical cancer incidence?
Author Response
|
1. Summary |
|
|
|
Thank you very much for taking the time to review this manuscript. Please find the detailed responses below and the corresponding revisions/corrections highlighted/in track changes in the re-submitted files
|
||
Comments
Q1 : Line 95-99, Was this the first HPV campaign in Togo? Was HPV vaccine available prior to Nov. 27, 2023? If so, how many girls aged 9-14 were vaccinated prior to this campaign?
R: Previously, Togo conducted a demonstration of the HPV vaccine introduction, covering a cohort of 15,272 girls aged 10 years who received two doses of HPV Vaccine in 2015 and 2016 in two health districts (Tchamba and Golfe).
Ministère de la santé du Togo : Rapport global. Projet de démonstration de la vaccination contre le virus du papillome humain pour la prévention et le contre le du cancer du col de l’utérus ; Togo - 2015 a 2017. Disponible sur : https://www.google.com/url?sa=t&source=web&rct=j&opi=89978449&url=https://www.pev.tg/download/rapport-global-2ans-projet-demo-vaccin-vph-cancer-col-togo-vf/&ved=2ahUKEwj4lJDPuJiMAxXWBfsDHeYCDDQQFnoECCwQAQ&usg=AOvVaw06EGIMdQvNgpM6XPgHe6aj
This has been added in the manuscript
Q2: Line 140, Please further clarify if fixed, outreach and mobile strategies were used in all three settings (i.e., school, health and community).
R: Yes, the 3 strategies were used in the all settings. We have provided more precision in the manuscript
Q3: Line 230, What might be the reasons that the HPV vaccine coverage rates were consistently higher for girls aged 9 compared to those aged 10-14 in all regions?
R: This could be explained by the fact that refusal were higher among 10 to 14 years old girls. Nine-year-old girls, most often are in primary school, faced less refusal than those older in secondary school. 10 to 14 years old girls might have more exposure to rumors on social media.
This has been added in the discussion
Q4: Line 243, What might be the reasons for high variation between districts within a region? For example, the vaccination rates vary from 37.7 to 90.1 in Plateaux.
R: The variation of vaccine coverage among districts within a region could be explained by the type of districts. Districts in rural setting have better vaccine coverage. Most of district with low coverage are closer to Maritime region that has high impact of misinformation circulating.
This has been added in the discussion
Q5: Line 268, Did cervical cancer burden vary greatly by geographic areas in Togo? How did vaccine coverage compare to the cervical cancer incidence in the area? Was the vaccine acceptance rate higher in the region/district with higher cervical cancer incidence?
R: Data on incidence of cervical cancer by region are not available in Togo.
However, it is noted that vaccination coverage during supplementary activities or in routine are higher from the nord region to the south
Reviewer 2 Report
Comments and Suggestions for Authors
See comments

Author Response
Thank you very much for taking the time to review this manuscript. Please find the detailed responses below and the corresponding revisions/corrections highlighted/in track changes in the re-submitted files
Comments
Тhe work conducted a study of vaccination of girls aged 9-14 years against HPV in all regions of the Republic of Togo. Regions with a high 104% vaccination rate (central administrative part) and zones with low efficiency - the southern regions - were identified. It has also been shown that vaccinations are more active in schools than outside of them. The age of those vaccinated was successfully chosen to be 9-14 years old, since at a later age vaccination is usually less effective, but still useful.
1.Of course, it is absolutely a pity that people (girls)often refuse to vaccinate.
In low social income countries, preventative cancer control is essential if there is no other option to combat the increasing burden of cancer, especially if the prevalence of cervical cancer is high and there are no options to overcome this problem.
R: Thank you for the valuable comments
2. In order to popularize vaccination with Gardasil, it should be pointed out that the antigenic proteins themselvesHPV6 L1, HPV11 L, HPV16 L1 and HPV18 L1 are practically harmless. They do not carry a carcinogenic load, but they are very stable and can be transporters of viral particles, since they themselves have motor properties.
R: This covered in the manuscript, in paragraph from line 80 to line 85
3. According to CLOBOCAN reports, the prevalence of HPV in the world is 60-100% now. But 60-100% of the population do not yet have cancer. It follows that the root cause of cancer is completely different life-threatening circumstances. For example, dirty water correlates with the risk of brain cancer (gliomas) and blood diseases (leukemia) in children.
R: Thank you for the comments.
We agree that additional factors such as immune response, co-infections, genetic predisposition…. influences in cancer development.
But, HPV remains a well-established cause of cervical and other anogenital cancers. Vaccination against HPV constitute a major preventable public health approach to reduce the burden of cervical cancers.
4. Side effects can only be caused by the presence of aluminum salts, erroneously named by manufacturers as adjuvants. Aluminum salts are toxic but cheap and are therefore used to support the three-dimensional structure of L1 antigenic proteins. If L1s lose the natural packaging of antigenic protein epitopes, then the vaccine will not induce the development of neutralizing antibodies.
R: We acknowledge the comment.
Of course, side effects were also monitored during the implementation of the campaign. We have provided the information on the number AEFI reported by region in Table 1 to complement the previous data provided in the manuscript
5. It makes no sense to be vaccinated with such a vaccine, lost due to lack of a “cold chain” or loss of natural configuration due to insufficiently careful storage since the safety of the natural configuration of the three-dimensional relative position of epitopes is the key to the success of vaccination.
R: Proper storage and handling of vaccines are indeed essential to maintaining their efficacy
6. Particular attention should be paid to vaccination in the coastal regions of the Republic of Togo. In mooring areas of industrial and private shipping, waste water is discharged directly into the water, incredibly polluting it. Coastal algae in surface waters are also depots for HPV and their reproduction.
R: We agree on these comments. Following the campaign, the vaccine was introduced in the routine. Several opportunities and strategies are being implemented to improve the vaccine coverage in Maritime and Grand Lome. These strategies include Big catch up initiative, and dialogues with communities Leaders
7.Likewise, sewage water is a reservoir for viruses and especially HPV, which have already lived for 400 billion years on earth and are accustomed to being stable in surviving difficult conditions.
R: Noted
8.Irrigation waters are also another source of HPV infection, and, what is dangerous, they are absorbed by agricultural plants and accumulate in them and people use these plants for food.
R: This is a great orientation for future research on HPV in Togo
9. Therefore, in order to make convincing propaganda in favor of vaccination, it is necessary first of all to give a fair assessment of the pollution of water, soil and plants. Then, in the harsh environment of an environmental disaster, parents will be able to convince their children to be more willing to get vaccinated, since vaccines are the safest means against the backdrop of a sharp deterioration in the environment, which contributes to the development of a cancer pandemic.
R: This guidance will foster our public health efforts on communication strategies concerning the benefits of vaccination.
Also, further studies on other factors of Cervical Cancer in Togo are important
10.According to the latest information on the number HPV types range from 330 types to 400 types, including vertebrates and invertebrates, insects, mollusks, snakes, turtles and even lichens. Differences can be up to 20% of nucleotide sequences
R: Thanks for the shared information